# Phase transitions of wave packet dynamics in disordered non-Hermitian systems

Hélène Spring[1*], Viktor Könye[2], Fabian A. Gerritsma[1], Ion Cosma Fulga[2], and Anton R. Akhmerov[1]

**1** Kavli Institute of Nanoscience, Delft University of Technology, P.O. Box 4056, 2600 GA Delft, The Netherlands
**2** Institute for Theoretical Solid State Physics, IFW Dresden and Würzburg-Dresden Cluster of Excellence ct.qmat, Helmholtzstr. 20, 01069 Dresden, Germany
*helene.spring@outlook.com

## Abstract

Disorder can localize the eigenstates of one-dimensional non-Hermitian systems, leading to an Anderson transition with a critical exponent of 1. We show that, due to the lack of energy conservation, the dynamics of individual, real-space wave packets follows a different behavior. Both transitions between localization and unidirectional amplification, as well as transitions between distinct propagating phases become possible. The critical exponent of the transition equals $1/2$ in propagating-propagating transitions.

## 1 Introduction

Wave propagation in a strongly disordered medium stops due to Anderson localization [1]. The latter depends only on macroscopic properties of the medium, such as its dimensionality, symmetries, and topological invariants. In one space dimension (1D), for instance, generic disorder will localize all eigenstates, even if the disorder strength is infinitesimally weak. On the other hand, weak anti-localization becomes possible in two- and higher-dimensional systems, depending on their symmetries [2]. In such cases, the full spectrum of a disordered energy-conserving medium contains regions of localized and extended states, which are separated by mobility edges.

Unlike energy-conserving media, non-Hermitian systems can exhibit fundamentally different behaviors in the presence of disorder. For instance, in the absence of energy conservation, it was found that weak disorder does not localize all states, even in 1D systems [3–5]. Instead, similar to their higher-dimensional Hermitian counterparts, in 1D non-Hermitian systems localized and delocalized eigenstates are separated by mobility edges across which the localization length diverges. A recent work has shown that this divergence is governed by a universal critical exponent taking the value $\nu = 1$ [6].

One of the practical uses of the theory of eigenstate localization is to predict the dynamics of individual wave packets. In Hermitian systems, this is straightforward: the initial wave packet is decomposed into a superposition of states with different energies. The wave packet components above the mobility edge diffuse through the medium, while those below the mobility edge stay localized. By contrast, non-Hermitian systems break energy conservation, such that it is no longer possible to directly describe the wave packet dynamics by separating it into components with different energies.

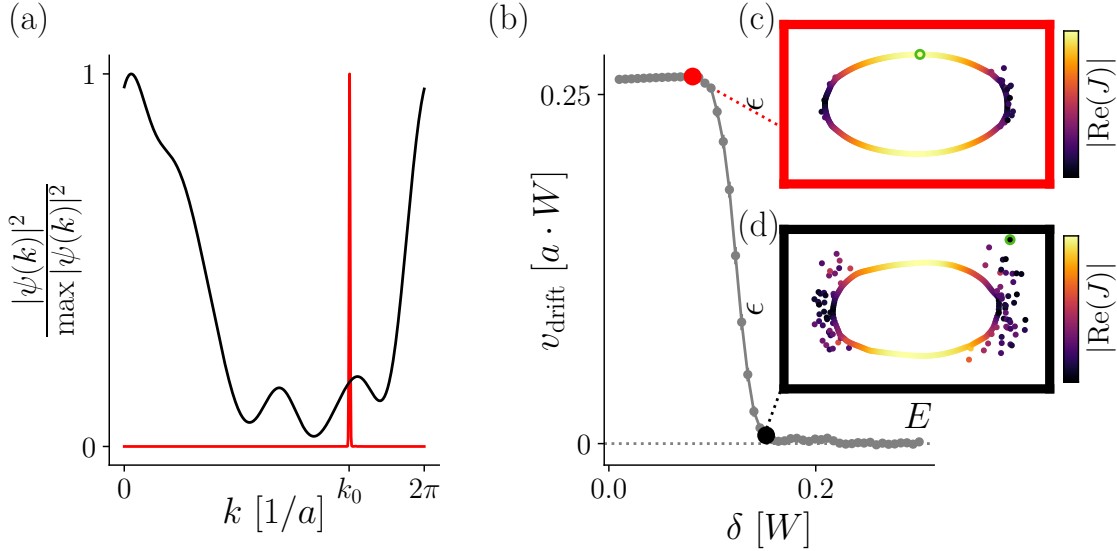

Figure 1: Maximally amplified waveforms of disordered Hatano-Nelson systems Eq. (1), with disorder $\delta$ in units of the model bandwidth $W$. (a) Magnitude of the Fourier components $|\psi(k)|^2$ of a wave packet evolved under $H_{HN}$ for $\delta = 0.01$ (red) and $\delta = 0.3$ (black). The maximally amplified Fourier component of the system with low disorder is marked by $k_0$. (b) The average drift velocity $v_{drift}$ as a function of disorder strength $\delta$, and $a$ the lattice constant. (c) Eigenvalues of $H_{HN}$ for a single disorder realization with disorder strength $\delta = 0.08$. The point of maximal amplification $\epsilon_{max}$ is highlighted with green. (d) Eigenvalues of a disordered system with disorder $\delta = 0.15$. $\epsilon_{max}$ is highlighted in green. Plot details in App. B.

Here we demonstrate that the difference between single energies and wave packets is profound. Because in the long-time limit any wave packet converges to a maximally amplified waveform, the asymptotic shape of the wave packet may change discontinuously when the system parameters are varied. This enables a direct transition between different unidirectionally amplified phases in addition to the previously known localization transition. Furthermore, in finite-size systems the fluctuations of the maximally amplified energy are self-averaging, which results in a critical exponent of $\nu = 1/2$.

The structure of the manuscript is as follows. In Sec. 2 we demonstrate the universal convergence of wave packets in weakly disordered systems to the maximally amplified waveform. In Sec. 3 we study the direct transition between distinct propagating phases. In Sec. 4 we show that the wave packet single-frequency transition differs from the static non-Hermitian single-frequency transition. We conclude in Sec. 5.

## 2 Maximally amplified wave packet

Unlike their Hermitian counterparts, one-dimensional (1D) non-Hermitian systems with no symmetries do not localize in the presence of weak disorder [3, 7, 8]. The different Fourier components of the wave packet, which are coupled by scattering events, are amplified at different rates, depending on the value of $\epsilon$, the imaginary part of their energy $E + i\epsilon$ [5]. The eigenstate whose eigenvalue has the largest positive imaginary component, $\epsilon_{max}$, is amplified the fastest. This means that any waveform in a weakly disordered medium converges to the maximally amplified waveform, forming an envelope in Fourier space

around the point of maximal amplification $k_0$.

To demonstrate this we consider a Hatano-Nelson Hamiltonian [3]:

$$H_{\mathrm{HN}} = \sum_j U_{0,j}|j\rangle\langle j| + \left(-\frac{W}{2}e^{-h} + U_{1,j}\right)|j\rangle\langle j+1|$$
$$+ \left(-\frac{W}{2}e^{h} + U_{2,j}\right)|j+1\rangle\langle j|, \tag{1}$$

where the sum runs over sites $j$ of the system, $W$ is a hopping parameter that sets the bandwidth of the system, $h$ fixes the degree of non-Hermiticity, and $U_{k,j}$ are the complex disorder coefficients whose real and imaginary parts are independently sampled from a normal distribution with zero mean and standard deviation $\delta_k$. Thus, $\delta_k$ models the strength of each type of disorder (onsite or hopping).

We time-evolve wave packets numerically by Taylor expanding the time-dependent Schrödinger equation to first order [See App. A for numerical methods]. For concreteness, throughout the following we consider an initial wave packet that has a Gaussian profile $u(x) = e^{-ikx}e^{-(x-x_0)/2\sigma^2}$. This wave packet is initialized at the center of the periodically wrapped lattice ($x_0 = 0$), with a width one tenth of the width of the lattice ($\sigma = L/10$) and with the same initial velocity ($k_x = \pi/2$, $k_y = 0$) for all simulations. The wave packet evolving under the weakly disordered Hatano-Nelson model Eq. (1) converges to an envelope around the point of maximal amplification $k_0$ [Fig. 1 (a), red curve]. For large disorder, the waveform acquires a non-universal shape whose center of mass is not guaranteed to be located around $k_0$ [Fig. 1 (a), black curve] .

The motion of the center of mass of the waveform in real space defines the drift velocity of the wave packet, $v_{\mathrm{drift}} = \partial_t\langle\psi|\hat{x}|\psi\rangle/\langle\psi|\psi\rangle$, with $\hat{x}$ the position operator. We evaluate this expression and obtain:

$$\partial_t\langle\psi|\hat{x}|\psi\rangle/\langle\psi|\psi\rangle = \frac{1}{2}\langle\psi|\partial_k\left(H + H^\dagger\right)|\psi\rangle$$
$$+ \frac{i}{2}\langle\psi|\{H - H^\dagger, \hat{x} - \langle\psi|\hat{x}|\psi\rangle\}|\psi\rangle, \tag{2}$$

where $\{\cdot, \cdot\}$ is the anti-commutator and where we normalize the wave function such that $\langle\psi|\psi\rangle = 1$.

The momentum-space non-Hermitian generalization of the current associated with a Hamiltonian $H$ is defined as $J(H) = -\partial_k H$. The first term of (2) is $\mathrm{Re}(\langle\psi|J|\psi\rangle)$ and for a single Bloch state $k_0$, $\partial_t\langle\psi|\hat{x}|\psi\rangle_{k_0} = \mathrm{Re}(J)|_{k_0}$. For the Hatano-Nelson Hamiltonian (1),

$$J(H_{\mathrm{HN}}) = \sum_j i\left(-\frac{W}{2}e^{-h} + U_{1,j}\right)|j\rangle\langle j+1|$$
$$+ i\left(\frac{W}{2}e^{h} - U_{2,j}\right)|j+1\rangle\langle j|. \tag{3}$$

At the localization transition, the drift velocity of the wave packet $v_{\mathrm{drift}}$ falls to 0 [Fig. 1 (b)]. We observe that below the localization transition, $v_{\mathrm{drift}}$ is finite and $\mathrm{Re}(J)$ at $\epsilon_{\max}$ is also finite [Fig. 1 (c)], and likewise when the wave packet is localized the $\mathrm{Re}(J)$ at $\epsilon_{\max}$ is 0 [Fig. 1 (d)].

Disorder shifts eigenvalues around in the complex plane, resulting in a different eigenstate becoming maximally amplified. Disorder also nontrivially changes the $\mathrm{Re}(J)$ of these eigenvalues. For strong disorder, the maximally amplified eigenstate is generically localized and $\mathrm{Re}(J) = 0$. The maximally amplified state may have nonzero $\mathrm{Re}(J)$ [Fig. 1 (c)], and therefore be delocalized [Fig. 1 (b)] or have zero $\mathrm{Re}(J)$ [Fig. 1 (d)], and therefore be

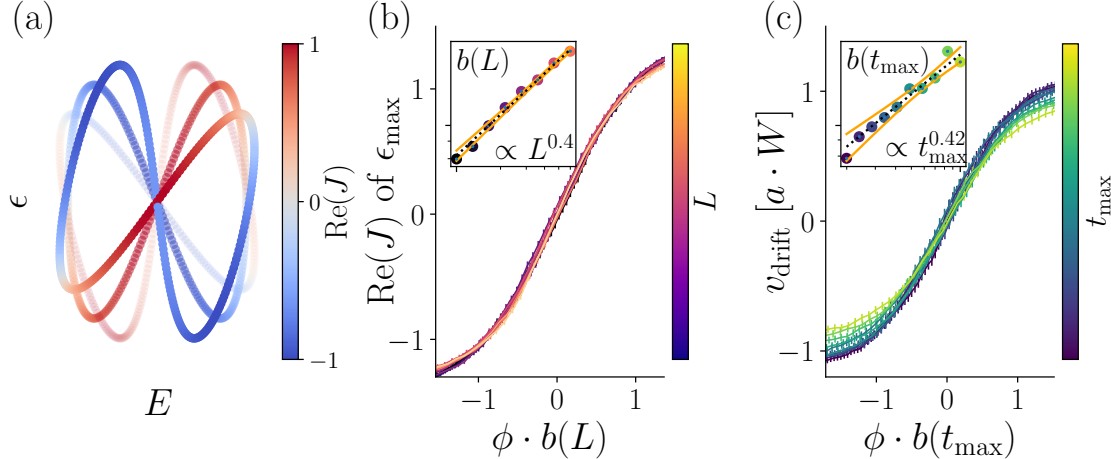

Figure 2:   Phase transition between left and right moving wave packets of Hamiltonian $H_8$ Eq. (4), with onsite and hopping disorder strength $\delta = 0.1$ in units of the system bandwidth $W$. (a) Real-space spectra for $\phi = 0.3$ (most transparent), $\phi = 0$ (intermediate transparency) and $\phi = -0.3$ (most opaque). (b) Rescaled $\mathrm{Re}(J)$ of the maximally amplified eigenstate and (c) $v_{\mathrm{drift}}$ around $\phi = 0$. Insets: scaling function of the slope at the transition and the 95% confidence interval. Plot details in App. B.

localized [Fig. 1 (b)]. If that state is delocalized, then the system delocalizes. Likewise if it is localized the system is localized even if other states in the systems are delocalized, since these states are always less amplified than the state at $\epsilon_{\mathrm{max}}$. Fig. 1 (d) shows that although delocalized states exist for $\epsilon < \epsilon_{\mathrm{max}}$, the system is localized because the maximally amplified state at $\epsilon_{\mathrm{max}}$ has $\mathrm{Re}(J) = 0$.

## 3   Direct transition between propagating phases

The expectation that propagating waveforms in non-Hermitian systems always evolve to the maximally amplified waveform suggests that a direct transition between competing propagating phases whose $\epsilon$ are close to $\epsilon_{\mathrm{max}}$ should be possible. Here we construct a Hamiltonian that hosts states propagating with opposite velocities at an $\epsilon$ close to $\epsilon_{\mathrm{max}}$:

$$
\begin{aligned}
H_8 = \sum_j U_{0,j}|j\rangle\langle j| &+ \left(\frac{We^{i\phi}}{2} + U_{1,j}\right)|j\rangle\langle j+1| \\
&+ \left(\frac{We^{i\phi}}{2} + U_{2,j}\right)|j+1\rangle\langle j| \\
&+ \left(\frac{We^{i\phi}}{2} + U_{3,j}\right)|j\rangle\langle j+2| \\
&+ \left(-\frac{We^{i\phi}}{2} + U_{4,j}\right)|j+2\rangle\langle j|,
\end{aligned}
\tag{4}
$$

where the sum runs over sites $j$ of the system, $W$ is a hopping parameter that sets the bandwidth of the system, $\phi$ rotates the spectrum in the complex plane, and where $\mathrm{std}(U_{k,j}) = \delta_k$

| Model | Quantity | Scaling exponent |
|---|---|---|
| $H_8$ (4) | $\mathrm{Re}(J)$ | $0.41 \pm 0.01$ |
| | $v_{\mathrm{drift}}$ | $0.45 \pm 0.03$ |
| $H_{\mathrm{HN}}$ (1) | $v_{\mathrm{drift}}$ | $0.38 \pm 0.04$ |

Table 1: Scaling parameters of the phase transitions shown in Fig. 2 and 3.

as in (1). The non-Hermitian generalization of the current $J$ is given by

$$
\begin{aligned}
J(H_8) = \sum_j & \; i\left(\frac{We^{i\phi}}{2} + U_{1,j}\right)|j\rangle\langle j+1| \\
& - i\left(\frac{We^{i\phi}}{2} + U_{2,j}\right)|j+1\rangle\langle j| \\
& + 2i\left(\frac{We^{i\phi}}{2} + U_{3,j}\right)|j\rangle\langle j+2| \\
& - 2i\left(\frac{We^{-i\phi}}{2} + U_{4,j}\right)|j+2\rangle\langle j|.
\end{aligned}
\tag{5}
$$

The spectrum of $H_8$ is composed of two lobes [Fig. 2 (a)]. The eigenstates associated to the eigenvalues at the top of the left lobe propagate to the left, and likewise those at the top of the right lobe propagate right, as shown by the sign of $\mathrm{Re}(J)$. By continuously tuning $\phi$ through 0, there is a discontinuous change in the eigenvalue with the largest positive imaginary component [Fig. 2 (a)] which leads to an abrupt transition between two different maximally amplified eigenstates. When $\phi \neq 0$, wave packets are amplified either predominantly to the left or to the right. The maximally amplified eigenstate of $H_8$ at $\phi = 0^-$ propagates to the left, and the one at $\phi = 0^+$ propagates to the right, meaning there is a metal-metal transition at $\phi = 0$. This transition is marked by a switch in the signs of both $\mathrm{Re}(J)$ and $v_{\mathrm{drift}}$ [Fig. 2 (b)-(c)].

In the presence of disorder and for finite system size, the average of $\mathrm{Re}(J)$ at $\epsilon_{\max}$ and $v_{\mathrm{drift}}$ changes linearly in the vicinity of $\phi = 0$, with an intermediate localized point at the middle of the transition. The slope of this transition increases with system size $L$ (for $\mathrm{Re}(J)$), and the total number of simulated time steps $t_{\max}$ (for $v_{\mathrm{drift}}$). We therefore confirm that the transition between the two propagating phases on either side of $\phi = 0$ does not go through a localized phase.

We examine finite-size scaling of the system at the transition. Due to the shape of the spectrum of $H_8$ [Fig. 2 (a)] on either side of the transition, the distribution of $E$ is bimodal, grouped around two values where $\epsilon$ is the largest. The variance of the individual peaks is the same at the transition point $\phi = 0$. Their standard deviations dictate the width of the transition, as $\phi \cdot t$ is required to be larger than these standard deviations in order for one part of the spectrum, and therefore one value of $\mathrm{Re}(J)$ to 'win' over the other.

There are several considerations we can make in order to estimate the scaling of these standard deviations as a function of system size. The variance of the peaks is equivalent to the variance of the expectation value of the disorder $U(x)$ in the system, $\mathrm{var}(\langle\psi|U(x)|\psi\rangle) = \mathrm{var}(\int_0^L \psi^*(x)U(x)\psi(x)dx)$. We reach an analytical expression for the scaling of the variance of the peak by considering that on either side of the transition, the

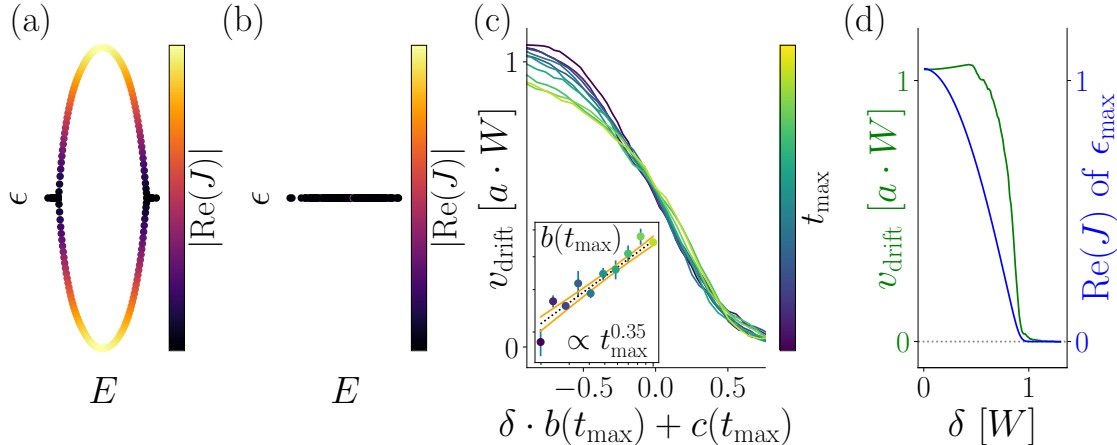

Figure 3: Finite-size scaling of the Hatano-Nelson Hamiltonian Eq. (1), with onsite disorder $\delta = \delta_0$ . (a)-(b) The Hatano-Nelson spectrum for disorder strength (a) $\delta = 0.3$ and (b) $\delta = 1.2$. (c) Rescaled wave packet drift $v_{\mathrm{drift}}$ at the transition point, collapsed using the relevant scaling parameter $b(t_{\mathrm{max}})$ and the irrelevant scaling parameter $c(t_{\mathrm{max}})$. Inset: fit of the scaling parameter $b(t_{\mathrm{max}})$ and the 95% confidence interval. (d) Comparison of $v_{\mathrm{drift}}$ (green) to $\mathrm{Re}(J)$ of $\epsilon_{\mathrm{max}}$ (blue) for system sizes $L = 10^3$. Plot details in App. B.

system contains delocalized phases that behave like plane waves and propagate throughout the system. The modulus of these propagating waves is approximately constant, $|\psi| \sim \mathrm{const}$. Therefore the dependence of the variance of the expectation value on system size $L$ is given by $\mathrm{var}(L^{-1} \int_0^L U(x)dx) = L^{-2} \cdot \mathrm{var}\left(\int_0^L U(x)dx\right) \propto L^{-2}L = L^{-1}$. The standard deviation of each peak of the distribution of $\epsilon$, and therefore the width of the transition, scales with $1/\sqrt{L}$. This leads to the expectation for the finite-size scaling of $b(L)$ to follow $\sqrt{L}$. This is in direct contrast to the expectation from single-energy studies where the critical exponent is $\nu = 1$ [6]. However, by construction the $H_8$ model transition is not a single-energy transition.

We fit $v_{\mathrm{drift}}$ and $\mathrm{Re}(J)$ of Fig. 2 with the function $a \tanh(b\phi)$, where $a$, $b$ are functions of system size $L$ for $\mathrm{Re}(J)$ fits, and functions of simulation time $t_{\mathrm{max}}$ for $v_{\mathrm{drift}}$. We choose $b$ as our relevant scaling parameter, since it measures the width of the transition. The numerical results for $\mathrm{Re}(J)$ at $\epsilon_{\mathrm{max}}$ show that the scaling is closer to $\nu = 1/2$ scaling than $\nu = 1$ scaling [see inset of Fig. 2 (c) and Table 1]. Although we have no analytical argument for the scaling of $v_{\mathrm{drift}}$, it also appears to follow $\nu = 1/2$ scaling [see inset of Fig. 2 (d) and Table 1]. App. C contains further discussion of the bimodal behavior.

## 4  Metal-insulator transition

The metal-metal transition behaves differently from the single-frequency response, which raises the question whether the metal-insulator transition is also different. In the presence of non-Hermitian disorder in both the onsite and hopping terms, the metal-insulator transition of the Hatano-Nelson Hamiltonian is the result of a discontinuous change in $\epsilon_{\mathrm{max}}$ [Fig. 1 (b)-(d)], and the same arguments as the metal-metal transition apply there. We therefore test whether a transition that does not involve a discontinuous switch of $\epsilon_{\mathrm{max}}$ and $E$ matches the single-frequency response. The original Hatano-Nelson Hamiltonian [3] fulfills this condition. We obtain this Hamiltonian by setting the disorder terms $\delta_i$ of Eq. (1) to be 0 except for $\delta_0$. Here the maximally amplified state is the last state to

localize, as the mobility edge moves from the largest absolute values of $E$ to the smallest [Fig. 3 (a)-(b)].

The shapes of the $v_{\mathrm{drift}}(\delta)$ curves of Fig. 3 do not lend themselves to a tanh fit. The scaling variable $b$ we choose in this case is the maximum slope during the transition. We also track an irrelevant scaling variable $c$ to ensure the superposition of the rescaled curves. The $v_{\mathrm{drift}}$ curves do not fully collapse at the transition [Fig. 3 (c)]. The scaling of $v_{\mathrm{drift}}$ is $b(t_{\mathrm{max}}) \propto t_{\mathrm{max}}^{0.38}$. We have no analytical expectation for the scaling of $v_{\mathrm{drift}}$, however we cannot rule out that the different critical exponent is due to finite-size effects and a finite resolution of the simulation, as demonstrated by the quality of the fit of the scaling parameter [see inset of Fig. 3 (c)]. Regardless of the actual value, the scaling behavior differs from $\nu = 1$.

Here $\mathrm{Re}(J)$ does not exhibit finite-size scaling and therefore does not show a phase transition. The phase transition of $v_{\mathrm{drift}}$ is thus ascribable to the non-linear term of Eq. (2) since it is absent from the linear term. $\mathrm{Re}(J)$ and $v_{\mathrm{drift}}$ nevertheless both fall to 0 at the same point [Fig. 3 (d)]. When taking the biorthogonal expectation value to calculate $\mathrm{Re}(J)$, finite-size scaling does occur [See App. D for discussion].

## 5   Conclusion

We showed that the dominant dynamics are attributable to a single point in the Fourier space of wave packets, which corresponds to the maximally amplified eigenstate. In the long time limit and in the presence of disorder, wave packets follow a behavior that is independent of initial conditions because they converge to the maximally amplified waveform. At the transition between distinct propagating phases, we found a $\nu = 1/2$ critical exponent. At the localization transition, the scaling also approaches $\nu = 1/2$. This clearly proves that wave packet transitions in disordered non-Hermitian media differ from the single-frequency response. Focusing on the metal-metal transition, we presented an analytical argument that proves that the value of $1/2$ is universal.

In our simulations we have observed that drift velocity of a wave packet $v_{\mathrm{drift}}(t_{\mathrm{max}})$ follows a scaling law similar to the scaling of an eigenstate in a finite system. It is not obvious that this equivalence is guaranteed, and further studies are required.

The nature of transitions in higher-dimensional non-Hermitian systems remains an open question. Preliminary results for two-dimensional systems show that the critical exponent differs from $\nu = 1/2$ [App. E]. It is therefore possible that the critical exponents of non-Hermitian systems are dimension-dependent.

Non-Hermitian systems are naturally realizable in experiment, and non-Hermitian wave packet dynamics are studied in photonic lattices and electrical circuits [9–11, 11–15]. The direct transition between propagating phases can be implemented as a switch tuned by a continuous parameter, with uses in control or sensor systems.

## Data availability

The data shown in the figures, as well as the code generating all of the data is available at [16].

## Author contributions

The initial project idea was formulated by I. C. F. and was discussed and later refined with contributions from H. S., V. K., and A. A. Results and code for non-Hermitian spectra were produced by H. S. The initial version of the wave packet propagation code and the error estimation were produced by V. K. Wave packet propagation with disorder was simulated by F. G. under the supervision of H. S. A. A related to relevant literature and formulated the question of comparing wave packet and single frequency phase transitions. The final data was generated by H. S. with input from I. C. F., A. A., and V. K. The scaling analysis was performed by H. S. with guidance from I. C. F. The manuscript was written by H. S. with contributions from I. C. F., A. A., and V. K. The project was managed by H. S. and A. A .

## Acknowledgments

The authors thank D. Varjas and M. Wimmer for their contributions to formulating the project idea and for helpful discussions. The authors thank Ulrike Nitzsche for technical assistance. H. S. thanks A. L. R. Manesco for helpful discussions and for reviewing the project code. A. A. and H. S. were supported by NWO VIDI grant 016.Vidi.189.180 and by the Netherlands Organization for Scientific Research (NWO/OCW) as part of the Frontiers of Nanoscience program. I. C. F. and V. K. acknowledge financial support from the DFG through the Würzburg-Dresden Cluster of Excellence on Complexity and Topology in Quantum Matter – ct.qmat (EXC 2147, project-id 390858490).

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

## A    Numerical methods

The time evolution of the wave packets was calculated using the scaled Taylor expansion method to first order [17–19], obtaining

$$|\psi(t + dt)\rangle = |\psi(t)\rangle - iH|\psi(t)\rangle dt, \tag{6}$$

where $|\psi(t)\rangle$ is the wave function at time $t$, $dt$ is the time step, and $H$ is the Hamiltonian dictating the time evolution. The simulation time $t$ and timesteps $dt$ are in units of the bandwidth $W$ of the system. We choose $dt = 0.01$, but we have separately checked that our results hold also for smaller time steps. In our simulations we initialize the system from the same real-space Gaussian wave packet. In order to ensure that the wave packets do not reach the system boundary, we limit the total number of time steps used for a simulation

to $t_{\max} = L/(a \cdot dt \cdot v_{\mathrm{drift}})$, with $L$ the system size, $a$ the lattice constant and $v_{\mathrm{drift}}$ the drift velocity of the wave packet for low disorder. Above the localization transition, $t_{\max}$ is not shortened in order to record instances of 'teleportation' of the drift center of the maximally amplified wave packet, which contribute to the average velocity.

The method is based on the following expression for the matrix exponential

$$\mathrm{e}^{-itH} = \lim_{N \to \infty} \left( I - \frac{itH}{N} \right)^N, \tag{7}$$

where $N$ is the number of time steps. Fixing the time step ($dt = t/N$) and the number of steps ($N$) we get an approximation for the time evolution operator as:

$$\mathrm{e}^{-itH} \approx (I - idtH)^N. \tag{8}$$

The error introduced at each subsequent time step can be estimated using the errors calculated for Taylor polynomials of the first order as [18, 19]:

$$\delta = \left\| \mathrm{e}^{-idtH} - I + idtH \right\| \leq \frac{dt^2 \|H\|^2}{2} \frac{1}{1 - \frac{dt\|H\|}{3}}, \tag{9}$$

where $\|\cdot\|$ is any well defined matrix norm, for simplicity we use the spectral norm. For normalized Hamiltonians $\|H\| = 1$ and $dt \leq 1$, the error introduced at each time step is $\delta \leq 3dt^2/4$.

## B  Model and plotting parameters

For Fig. 1 (b), $\delta$ is varied between 0.01 and 0.3 in 50 steps, and the average drift velocity is averaged over 600 different disorder configurations. The spectra of panels (c) and (d) are calculated for systems composed of 300 lattice sites, and parameter $h$ set to 0.3. For panels (a) and (b), the wave packet evolution was performed on system sizes of 600 sites, in steps of $dt = 0.01$ for 60000 steps. For panel (a) the results displayed in the figure are taken at the last step of the time evolution. The disorder strength $\delta$ is given in units of $W$ the bandwidth of $H_{\mathrm{HN}}$.

For Fig. 2, the spectra, $\mathrm{Re}(J)$ and the wave packet results are obtained for systems with sizes $L \in \{199, 238, 285, 341, 408, 488, 584, 698, 836, 1000\}$. Results for $\mathrm{Re}(J)$ and wave packets are averaged over 2000 and 500 different disorder configurations respectively. The wave packet evolution was performed in steps of $dt = 0.01$ for $L/dt$ steps. The tilt angle $\phi$ was varied between $-0.1$ and $0.1$ in the following way: 20 points between $-0.1$ and $-0.03$, 100 points between $-0.03$ and $0.03$, and 20 between $0.03$ and $0.1$. The disorder strength is set to $\delta = 0.1$ in units of the bandwidth $W$ of the Hamiltonian Eq. (4).

For Fig. 3, the parameter $h$ is set to 0.3. Ten different system sizes $L$ are simulated, $L \in \{199, 238, 285, 341, 408, 488, 584, 698, 836, 1000\}$. Results are averaged over 500 different disorder configurations for each value of disorder strength. The wave packet evolution was performed in steps of $dt = 0.01$ for $L/dt$ steps, with the values of $L$ as stated above.

For the insets of Fig. 2 (b)-(c) and Fig. 3 (c), the error of the scaling fit is shown using the 95% confidence interval.

For Fig. 4, panel (a) data is made up of 5000 disorder configurations, for systems 800 sites long. Panel (c) data is made up of 3000 disorder configurations, for systems 800 sites long. Panel (c) data is made up of 2000 disorder configurations, for systems 1000 sites long. Panel (d) data is made up of 3000 disorder configurations, for systems 800 sites long

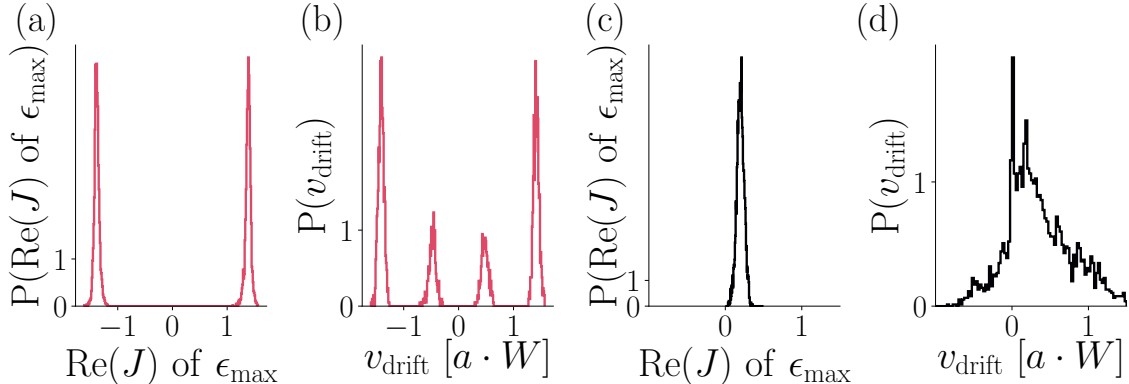

Figure 4: Multimodal and bimodal distributions of $\mathrm{Re}(J)$ and $v_{\mathrm{drift}}$ of the Hamiltonians $H_8$ [Eq. (4)] and $H_{\mathrm{HN}}$ [Eq. (1)]. (a), (c) distributions of $\mathrm{Re}(J)$ of the maximally amplified state and $v_{\mathrm{drift}}$ of the $H_8$ model at $\phi = 0$ and $\delta = 0.1$. (b), (d) distributions of $\mathrm{Re}(J)$ of the maximally amplified state and $v_{\mathrm{drift}}$ of the $H_{\mathrm{HN}}$ model at $\phi = 0$ and $\delta = 0.8$. Plot details in App. B.

and $\delta = 0.8$. The wave packet results are obtained for time evolution step size $dt = 0.01$ and total time steps $L/dt$.

Fig. 7 is composed of unscaled data that was obtained and used in Fig. 2 and Fig. 3.

For Fig. 6, the wave packet evolution was performed in steps of $dt = 0.01$ for $L/dt$ steps. Five different system sizes $L \times L$ were simulated, with $L \in \{64, 85, 113, 150, 199\}$. Disorder strengths were varied between 0.01 and 0.5 in 50 steps. For each disorder strength, the result is averaged over 400 different disorder configurations.

For Fig. 5, panel (a) data for $\delta = 0.01$ is composed of 100 different disorder configurations for systems of 800 sites. Panel (a) data for $\delta = 0.1$ is made up of 1000 disorder configurations, for systems 1000 sites long. Panel (b) data is made up of 5000 disorder configurations, for systems 800 sites long and $\delta = 0.8$. For panel (c), results for $\mathrm{Re}(J)$ and wave packets are averaged over 500 different disorder configurations. The tilt angle $\phi$ was varied between $-0.1$ and $0.1$ in the following way: 20 points between $-0.1$ and $-0.03$, 100 points between $-0.03$ and $0.03$, and 20 between $0.03$ and $0.1$. Results are obtained for systems with sizes $L \in \{199, 238, 285, 341, 408, 488, 584, 698, 836, 1000\}$. The disorder strength is set to $\delta = 0.1$ in units of the bandwidth $W$ of the Hamiltonian (4). For the insets of panels (c)-(d), the error of the scaling fit is shown using the 95% confidence interval.

## C   Multimodal behavior

Here we discuss the shape of the distributions of $\mathrm{Re}(J)$ and $v_{\mathrm{drift}}$ of both the $H_8$ [Eq. (4)] and $H_{\mathrm{HN}}$ [Eq. (1)] models around the transition point.

For $H_8$, the distribution of $\mathrm{Re}(J)$ of the maximally amplified eigenstate is bimodal [Fig. 4 (a)]. The distribution of $v_{\mathrm{drift}}$ is multimodal [Fig. 4 (c)]. The multimodality arises from the disorder nontrivially shifting eigenvalues of $H_8$ in the complex plane, creating two bimodal distributions for $v_{\mathrm{drift}}$, one on each side of the transition in $\phi$. The same multimodal behavior is seen in $\mathrm{Re}(J)$ of the maximally amplified eigenstate when using biorthogonal expectation values to calculate $J$ [App. D, Fig. 5 (a)].

For $H_{\mathrm{HN}}$, $\mathrm{Re}(J)$ does not exhibit a transition [Fig. 3 (d)], and its distribution close to the $v_{\mathrm{drift}}$ transition is centered around a small but finite value [Fig. 4 (b)]. The scaling of

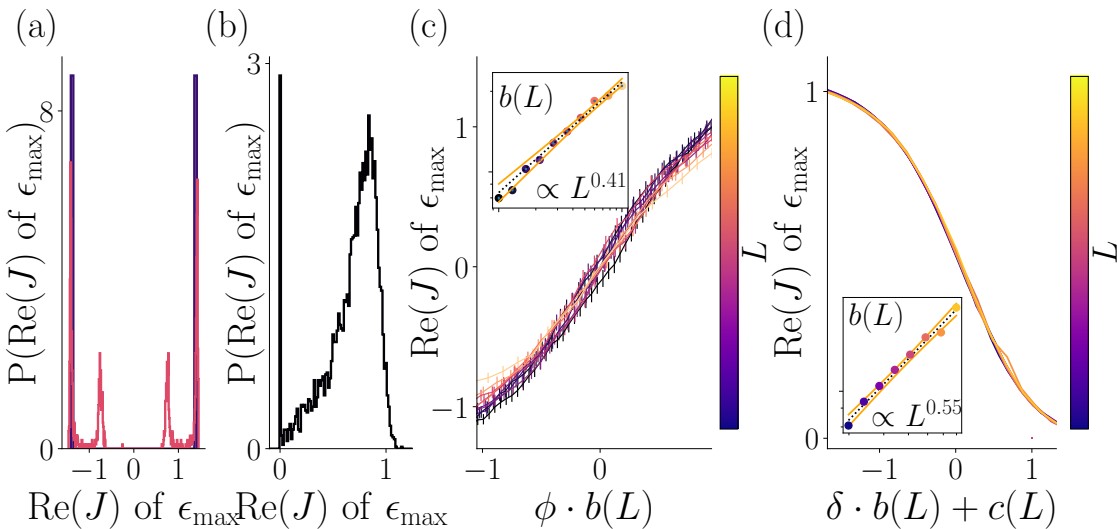

Figure 5: Re$(J)$ results using biorthogonal expectation values for Hamiltonians $H_8$ [Eq. (4)] and $H_{\mathrm{HN}}$ [Eq. (1)]. (a) distributions of Re$(J)$ of the maximally amplified state of the $H_8$ model at $\phi = 0$ and $\delta = 0.1$ and $\delta = 0.01$ in units of the bandwidth $W$. (b) distribution of Re$(J)$ of the maximally amplified state of the $H_{\mathrm{HN}}$ model at $\delta = 0.8$. (c) Rescaled Re$(J)$ of the maximally amplified state of the $H_8$ model for $\delta = 0.1$. (d) Rescaled Re$(J)$ of the maximally amplified state of the $H_{\mathrm{HN}}$ model for $\delta = 0.1$. Insets of (c) and (d): scaling functions of the slope at the transition and the 95% confidence interval. Plot details in App. B.

$v_{\mathrm{drift}}$ of the Hatano-Nelson model $H_{\mathrm{HN}}$ does not exactly follow $\sqrt{t_{\max}}$ [Table 1 and Fig. 3 (d)] but a bimodal distribution is still observed close to the transition [Fig. 4 (d)]. Close to the transition point, the distribution of $v_{\mathrm{drift}}$ has two peaks, with one broad peak centered around a finite value, and the other delta function peak around 0. The $v_{\mathrm{drift}}$ around 0 originates from disorder configurations that result in localization, and the $v_{\mathrm{drift}}$ with finite velocity originates from disorder configurations where propagation is still possible.

# D    Biorthogonal expectation value

In the results of the manuscript, we calculated Re$(J)$ of the state $m$ as Re$(\langle\psi_m|J|\psi_m\rangle)$ such that $\langle\psi_m| = |\psi_m\rangle^\dagger$. In this section we calculate Re$(J)$ of state $m$ as Re$(\langle\psi_m|J|\psi_m\rangle)$ such that $\langle\psi_m| = |\psi_m\rangle^{-1}$, that is to say $\langle\psi_m|$ is the m-th left eigenstate and $|\psi_m\rangle$ is the m-th right eigenstate. We refer to this Re$(\langle\psi_m|J|\psi_m\rangle)$ as the biorthogonal expectation value of Re$(J)$. The behavior of Re$(J)$ is significantly impacted by this change in expectation value, as shown in Fig. 5.

For the $H_8$ model, similarly to Fig. 4 for low disorder ($\delta = 0.01$) the distribution of Re$(J)$ of the maximally amplified eigenstate is bimodal [Fig. 5 (a)]. At finite disorder, the distribution becomes multimodal, similar to $v_{\mathrm{drift}}$ [Fig. 4 (c)]. The scaling parameter at the transition of the biorthogonally projected Re$(J)$ scales as $L^{0.44\pm0.01}$ [Fig. 5 (c)].

The Hatano-Nelson model $H_{\mathrm{HN}}$ also exhibits bimodal behavior [Fig. 5 (b)], similarly to the distribution of $v_{\mathrm{drift}}$ [Fig. 4 (d)]. Close to the transition point, the distribution of Re$(J)$ has two peaks, with one broad peak around the low-disorder Re$(J)$ value 1.1, and the other delta function peak around the high disorder Re$(J)$ value 0. In the biorthogonal case, the Re$(J)$ of the $H_{\mathrm{HN}}$ displays a phase transition. The scaling of the transition width

| Parameter | value |
|:---:|:---:|
| $t_{x,+}$ | 1 |
| $t'_{x,+}$ | 0 |
| $t_{x,-}$ | 0.8 |
| $t'_{x,-}$ | 0 |
| $t_{y,+}$ | 0 |
| $t'_{y,+}$ | 0 |
| $t_{y,-}$ | 0 |
| $t'_{y,-}$ | 1 |

Table 2: Parameters used for simulating Hamiltonian Eq. (10).

is found to scale close to $\sqrt{L}$, as $L^{0.55\pm0.02}$ [Fig. 5 (d)], similarly to the $H_8$ case.

The $\mathrm{Re}(J)$ calculated using the biorthogonal expectation value appears to follow the behavior of $v_{\mathrm{drift}}$ more closely, but we do not have an argument as to why this would be the case.

# E    Results in two dimensions

We consider the following two-dimensional non-Hermitian model:

$$
\begin{aligned}
H_N &= \sum_{d=1}^{N} H_d, \\
H_d &= \sum_{j}^{L_d} \left( t_{x_d,+} + it'_{x_d,+} \right) |x_{d,j+1}\rangle\langle x_{d,j}| \\
&+ \left( t_{x_d,-} + it'_{x_d,-} \right) |x_{d,j}\rangle\langle x_{d,j+1}|,
\end{aligned}
\tag{10}
$$

where the sum runs over all the lattice sites $j$ and the spatial dimensions $d$ of a $N$-dimensional system with $L/a = \frac{1}{a}\sum_{d}^{N} L_d$ sites, with $a$ the lattice constant. $x_d$ corresponds to the spatial coordinate in dimension $d$. We choose $N = 2$.

The parameters we use in simulating this model are found in Table 2, and yield the spectrum shown in Fig. 6 (a)-(b).

We fit the function $a\tanh(b\delta + c)$ to the localization transition of $v_{\mathrm{drift}}$ as a function of $\delta$, and extract $b(t_{\max})$. $b(t_{\max})$ scales as $t_{\max}^{0.63\pm0.05}$ [Fig. 6 (d)]. The critical exponent of 2D non-Hermitian dynamic systems approaches $\nu = 0.5$. However it is not possible from these results to say whether the critical exponent of non-Hermitian systems is dimension-dependent or not.

# F    Unscaled results

The results for $\mathrm{Re}(J)$ at $\epsilon_{\max}$ and $v_{\mathrm{drift}}$ shown in Fig. 2 and Fig. 3 are rescaled by the scaling variables $b$ (and $c$ in the case of $v_{\mathrm{drift}}$). Fig. 7 contains the unscaled data used to obtain Fig. 2 and Fig. 3, as well as the rescaled data for comparison.

We do not show rescaling of the $\mathrm{Re}(J)$ at $\epsilon_{\max}$ curves of Fig. 7 (c1), since they do not exhibit scaling.

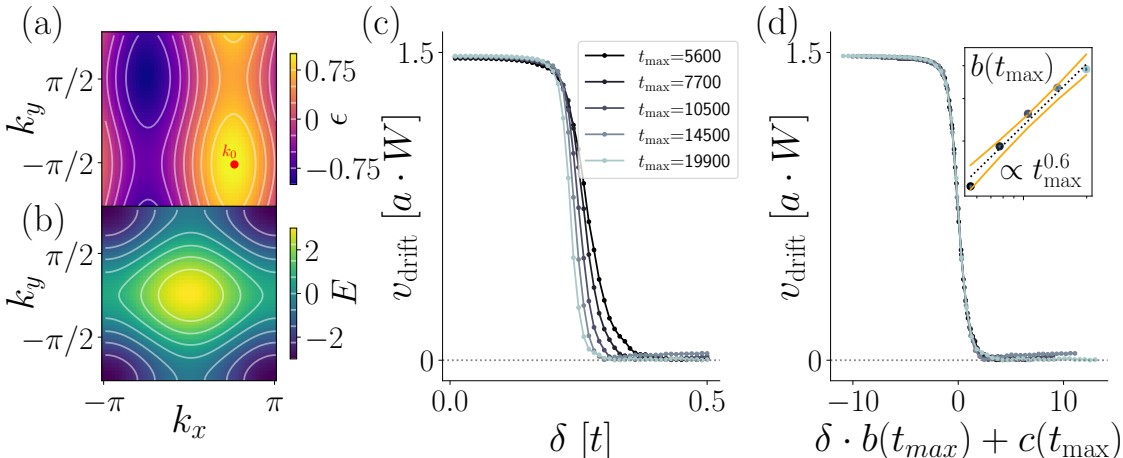

Figure 6: Finite-time scaling of two-dimensional non-Hermitian model Eq. (10) with parameters 2. (a)-(b) The Brillouin zone of (a) the imaginary part of the energy $\epsilon$ and (b) the real part of the energy. (c) The unscaled localization transition of $v_{\text{drift}}$ as a function of $\delta$. (d) The rescaled curves of (c). Inset: scaling of the sharpness of the transition. Plot details in App. B.

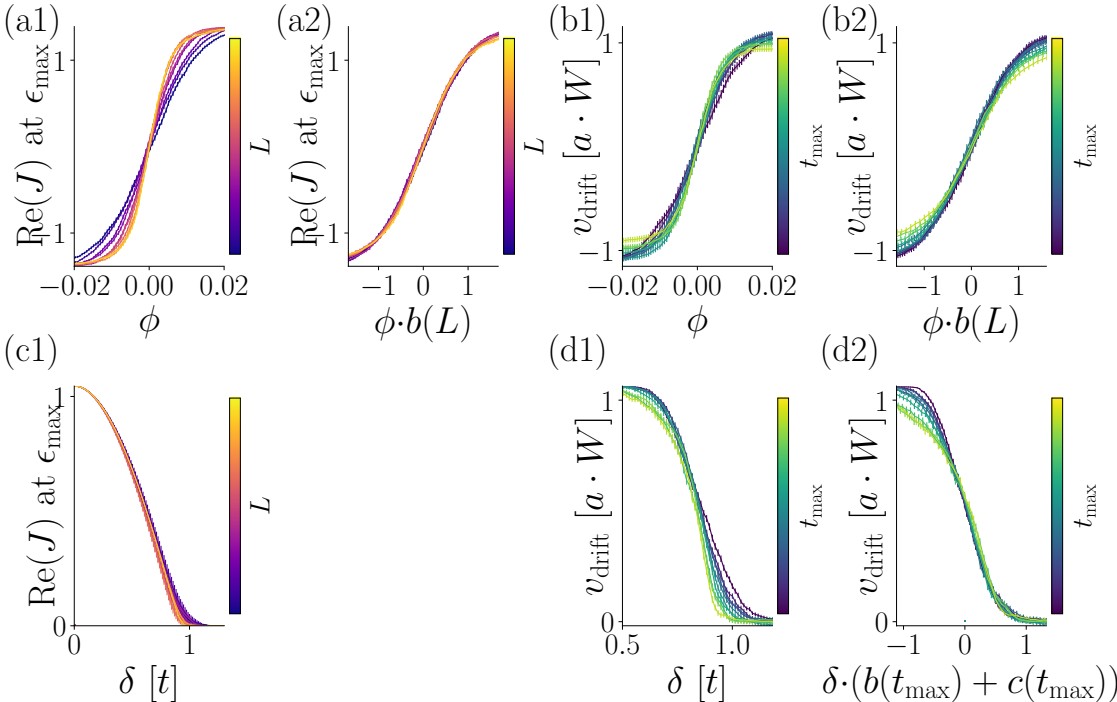

Figure 7: Unscaled (a1,b1,c1,d1) and rescaled (a2,b2,d2) $\text{Re}(J)$ at the point of maximal amplification $\epsilon_{\text{max}}$ and $v_{\text{drift}}$ at the transition point. (a1)-(b2) Results for the $H_8$ model Eq. (4) used in Fig. 2. (c1)-(d2) Results for the $H_{\text{HN}}$ model Eq. (1) used in Fig. 3. Plot details in App. B.