# Peer review of "Phase transitions of wave packet dynamics in disordered non-Hermitian systems"

_SciPost Physics_

## Round 1 · Referee Report · Anonymous (Referee 2) · 2023-2-23

Report

The authors studied the wave packet dynamics in random non-Hermitian systems, where they found localization-unidirectional amplification transition and transition between propagating phases. I found the theme of the paper interesting, but the readers of this paper will not be able to reproduce the results, since some of the information for the numerical calculations are not provided in the paper. I also have several comments on the results.
1. One of the important finding of this paper is the scaling behaviors of v_drift and Re(J) (Figs. 2 (b)(c) and 3(c)). But I don’t see the scaling equation in the text. The physical quantity such as v_drift is a function of several system parameters like L, delta_0, delta_1,…,h, phi, and the scaling form should be v_drift/(a W)=f(a, L, delta_0, delta_1,…,h, phi,..)= F(x(a, delta_0,…)L^1/nu) where x is the relevant scaling variable. Is this what the authors mean? What is the relation between this x and b in the paper?
2. The scaling exponent nu=1/2 is outside the confidence intervals in Table 1. Do the authors think the exponent is 1/2 as they write in the abstract and Conclusion? If it is the case, the authors should explain why 1/2 is outside the confidence intervals of Table 1.
3. Below Eq. (1), the authors write that they set the initial velocity as k_x=pi/2 and k_y=0. What is k_y? I think the authors are considering one dimensional system.
4. In the caption of Fig. 1, the value of delta is written as 0.01 and 0.03. What is this delta? delta_i (i=0,1,2,3 and 4) are defined in Eqs. (1) and (4), but not delta.
5. I also think the values of h characterizing the asymmetry of hopping should be given.
6. The authors write that h fixes the degree of non-Hermiticity. But I think the difference between U_1,j and U_2,j is also the origin of non-Hermiticity.
7. The authors used Taylor expansion to follow the dynamics. Isn’t Chebyshev expansion better? Or is there a difficulty in applying Chebyshev expansion for non-Hermitian systems? If it is the latter case, the authors should comment on this.
8. In the definition of the drift velocity, v_dfrit=partial_t <psi|x|psi>/<psi|psi>, the authors should clarify if the time derivative operates on the normalization factor <psi|psi> or not.
9. I have difficulty understanding Fig. 2(a), which the authors call “real-space spectra”. Are the axes real and imaginary parts of eigen energy? What is the definition of epsilon and E in this figure?
10. Why do we call the Hamiltonian in Eq. (4) as H_8?
  • validity: -
  • significance: -
  • originality: -
  • clarity: -
  • formatting: -
  • grammar: -

Author:  Helene Spring  on 2024-02-26  [id 4325]

(in reply to Report 2 on 2023-02-23)
Category:
answer to question

Dear referee,
We attach the full response and the redline manuscript with changes highlighted.
Best regards,
The authors

Attachment:

combined_UNKxJ1b.pdf

---

## Round 1 · Referee Report · Anonymous (Referee 1) · 2023-2-23

Strengths

  1. The manuscript finds a new type of dynamical phase transitions in non-Hermitian disordered systems.

Weaknesses

  1. One of the main claims, the critical exponent $\nu = 1/2$, is inconsistent with the numerical results.

  2. The results are mainly based on some specific models, and their generality and universality are unclear.

Report

The authors study the wave packet dynamics of non-Hermitian disordered systems in one dimension. They find a phase transition of the drift velocity and current defined from the wave packet dynamics in a couple of models. They show the critical exponents different from those for the localization length and conductance in some previous works and further clarify the different nature of the phase transitions.

Recently, the physics of non-Hermitian systems has attracted considerable interest in both theory and experiments. The role of disorder and concomitant localization transitions in non-Hermitian systems has also been actively studied. In this context, I believe that this manuscript, which finds a new type of dynamical phase transitions in non-Hermitian disordered systems, should make a significant contribution in non-Hermitian physics.

However, this manuscript contains several unclear descriptions, which preclude me from recommending the publication at this stage. For the publication of this manuscript in SciPost Physics, I would like to request that the authors thoroughly address the following concerns.

Requested changes

[Major comments]

(I) The authors claim that the obtained critical exponent for the current and drift velocity is 1/2. For example, the abstract reads, “The critical exponent of the transition equals 1/2 in propagating-propagating transitions”. Additionally, in Sec. 5, the manuscript reads, “At the transition between distinct propagating phases, we found a $\nu=1/2$ critical exponent. At the localization transition, the scaling also approaches $\nu=1/2$”. However, all the numerically obtained critical exponents summarized in Table I significantly deviate from $\nu=1/2$. Similarly, in the last paragraph of Sec. 3, the manuscript reads, “Although we have no analytical argument for the scaling of $v_{\mathrm{drift}}$, it also approaches to follow $\nu=1/2$ scaling”. However, I do not believe that the authors’ numerical results approach $\nu=1/2$ within the error bars, and do not find these explanations convincing. On the basis of the present numerical results, I believe that the statement that the “The critical exponent of the transition equals 1/2” is scientifically wrong. The authors should clarify this point.

I suspect that the authors’ expectation of $\nu=1/2$ is due to the implicit assumption that this critical exponent should be universal (i.e., do not depend on specific details of systems but solely on fundamental properties such as symmetry and dimension) and be simple in one dimension. However, the universality of the authors’ critical exponent $\nu$ is unclear only from the present results, in contrast to the conventional critical exponents that universally characterize the Anderson transitions (please see also below). Thus, I cannot rule out the possibility that the authors’ critical exponent is different even in the same symmetry class and spatial dimension, and is not a simple rational number even in one dimension; I believe that it is significant to make a correct statement, rather than an intuitive but naive speculation.

(II) I fail to clearly understand how general and universal the authors’ results are, in contrast to the conventional quantities whose critical exponents universally characterize the Anderson transitions (e.g., localization length, conductance). In particular, it is unclear whether we have a single critical exponent in the same symmetry class and spatial dimension. My concern here is partially because the authors’ phase transition of the wave packet dynamics depends only on a few complex-valued eigenvalues that have the largest imaginary part, which contrasts with the conventional Anderson transitions that arise from the collective behavior of quasiparticles. For example, in the last paragraph of Sec. 4, the manuscript reads, “Here $\mathrm{Re} \left( J \right)$ does not exhibit finite-size scaling and therefore does not show a phase transition”. This implies that the nature of phase transitions depends on specific details of models even in the same symmetry class and spatial dimension. Furthermore, in the first paragraph of Sec. 5, the manuscript reads, “Focusing on the metal-metal transition, we presented an analytical argument that proves that the value of 1/2 is universal”. Although I agree that the authors provide an analytical argument, it still assumes some details of models that are not specified solely by symmetry and dimension, and I do not find the argument fully convincing. While the comprehensive discussions may go beyond the aim and scope of the present manuscript, I would like to request that the authors elaborate more on the universality of their results.

(III) While the authors’ results [e.g., Fig. 1(b)] clearly show a phase transition as a consequence of the competition between non-Hermiticity and disorder, I fail to clearly understand whether this phase transition is continuous or discontinuous. Correspondingly, while the authors numerically obtain the critical exponents, I cannot exclude the possibility that this phase transition is actually discontinuous and that the obtained critical exponents are due to the finite-size effect and not well defined in the infinite-size limit. While some of the authors’ results may already support the continuous phase transitions, I cannot clearly understand it in the present manuscript. Thus, I would like to request that the authors clearly demonstrate that the phase transitions observed in this manuscript are indeed continuous and are characterized by well-defined critical exponents.

[Minor (specific) comments]

(i) In the third paragraph of Sec. 2, the initial velocity is specified by “$k_x = \pi/2, k_y = 0$”. I find this condition unclear simply because the two momenta $k_x$ and $k_y$ are given even in one dimension. The authors should clarify this point. While $k_y$ may be relevant to the calculations of the two-dimensional model in Appendix E, it should not be written here since it is confusing.

(ii) In Eq. (2), the authors provide the equation of motion for the wave packet dynamics in non-Hermitian Hamiltonians. I find some of the descriptions ambiguous and unclear. First, while the drift velocity is defined as “$\partial_t \langle \psi | \hat{x} | \psi \rangle/\langle \psi | \psi \rangle$”, it is unclear whether this means “$\left[ \partial_t \langle \psi | \hat{x} | \psi \rangle \right]/\langle \psi | \psi \rangle$” or “$\partial_t \left[ \langle \psi | \hat{x} | \psi \rangle/\langle \psi | \psi \rangle \right]$” (I suspect that the latter seems reasonable, though). Additionally, while the manuscript reads “… where we normalize the wave function such that $\langle \psi | \psi \rangle=1$” just after Eq. (2), it is unclear when this normalization is introduced since it does not seem to be introduced before Eq. (2). Since the norm of wave functions is time dependent for non-Hermitian Hamiltonians, the precise way of normalization affects the way of the time derivative $\partial_t$ and should be important. Correspondingly, I think that it would be better to provide a more detailed explanation on the derivation of Eq. (2).

(iii) In the fifth paragraph of Sec. 2, the current for non-Hermitian Hamiltonians is introduced. However, the validity of this definition is unclear. More specifically, the authors define the current operator $J$ by $J = - \partial_k H$. While this definition is arguably reasonable for Hermitian Hamiltonians $H$, its validity for non-Hermitian Hamiltonians is unclear. Correspondingly, for non-Hermitian Hamiltonians $H$, this current operator can be non-Hermitian, as is indeed the case for Eq. (3). While the authors only focus on the real part of the non-Hermitian current operator, they clarify the meaning of the imaginary part.

(iv) In Sec. 3, I fail to clearly understand the precise meaning of “direct” in the terminology “direct transition”, which the authors should clarify.

  • validity: low
  • significance: good
  • originality: good
  • clarity: low
  • formatting: reasonable
  • grammar: reasonable

Author:  Helene Spring  on 2024-02-26  [id 4326]

(in reply to Report 1 on 2023-02-23)
Category:
answer to question
reply to objection

Dear referee,
We attach the full response and the redline manuscript with changes highlighted.
Best regards,
The authors

Attachment:

combined_TewXM4f.pdf

---

## Round 1 · Referee Report · Anonymous (Referee 3) · 2023-2-27

Strengths

  1. The paper is well-written.

  2. Both the setup and the results presented clearly.

Weaknesses

  1. Narrow focus on a very special class of models.

Report

This paper describes a theoretical analysis of the localization transition in non-Hermitian lattices. Using a couple of specific models, it uncovers the processes by which the transition from a localizing to a non-localizing phase sets in for these lattices, which appears to be qualitatively different from the standard metal-to-insulator transition of Hermitian models. Essentially, any arbitrary initial wavepacket will evolve into the lattice's maximally amplified mode---the one with the largest value of Im(E)---so whether or not localization sets in depends on the dynamical characteristics of this maximally amplified mode. As disorder increases, it is possible for a mode with different characteristics to take over as the maximally amplified mode, giving rise to a localization transition.

One concern I have is that the introduction and conclusion refer to non-Hermitian systems as a general class, with the implication that these findings apply to other non-Hermitian systems, at least in 1D. Yet the study is based on the 1D Hatano-Nelson (HN) model, and other models with "point gaps" (spectra that form one or more loops in the complex energy plane), with periodic boundary conditions. As the authors are aware, the HN model and its variants have rather special properties not present in other non-Hermitian models. For one thing, the HN model's non-Hermitian hoppings bias propagation along one direction; for another, lattices with point gaps exhibit a "non-Hermitian skin effect" causing the periodic and finite lattice have extremely different eigenstates.

Thus, even if the conclusions of the study itself are sound, it doesn't seem warranted to confidently draw lessons for other kinds of non-Hermitian systems. Note also that the class of non-Hermitian lattices most easily realized in photonics or electrical circuits (as mentioned in the conclusion) is those with on-site loss and no point gaps.

Within the context of the HN model and its relatives, I find it strange that the authors provide scarcely any physically-motivated discussion of the model's characteristics (e.g., the disorder-free HN model's propagation direction bias), and how they relate to the findings. For instance, in Fig. 1(c) it seems evident that the loop in the complex energy plane is continuable to the spectrum of the disorder-free HN model (roughly, waves moving along/opposite the bias direction get amplified/damped). With more disorder, the most affected eigenstates, whose energies migrate furthest away from the loop, are naturally those at the band extrema.

There are a couple of more minor issues with the figures, which will be detailed in the next section.

If these requests can be satisfactorily accommodated, the paper can be accepted.

Requested changes

  1. Either the introduction and conclusion should be toned down, or the authors should explain why their results ought to generalize.

  2. Given the recent interest in point gaps, the non-Hermitian skin effect, etc., it would also be good if the authors could explicitly comment about whether their findings --- particularly the route to localization transition via the "maximally amplified wave packet" mechanism --- should hold for non-Hermitian systems that don't have point gaps, and/or those with open boundary conditions.

  3. Consider providing more discussion of how the disorder-free HN model properties lead to the localization transition, as mentioned above.

  4. Fix the following issues with the figures:

  5. Below Eq. (1), the authors define multiple disorder parameters, denoted by delta_k. However, Fig. 1 refers to a single disorder parameter, delta, with no accompanying explanation. The other figures have a similar issue.

  6. In Fig. 1(c)--(d), the axes should include ticks for at least the zero point.

  7. In Fig. 1, it would be nice if the authors could "close the loop" of the argument by showing that the largest-ϵ state indeed produces the time-domain results shown in (a).

  • validity: high
  • significance: good
  • originality: good
  • clarity: high
  • formatting: excellent
  • grammar: perfect

Author:  Helene Spring  on 2024-02-26  [id 4324]

(in reply to Report 3 on 2023-02-27)
Category:
answer to question
reply to objection

Dear referee,
We attach the full response and the redline manuscript with changes highlighted.
Best regards,
The authors

Attachment:

combined.pdf

---

## Editorial Decision

resubmitted